# Optimization by Central Composite Experimental Design of the Synthesis of Physically Crosslinked Chitosan Spheres

**DOI:** 10.3390/biomimetics5040063

**Published:** 2020-11-20

**Authors:** Sara Isabel Zamora Lagos, Jefferson Murillo Salas, Mayra Eliana Valencia Zapata, José Herminsul Mina Hernández, Carlos David Grande Tovar

**Affiliations:** 1Escuela de Ingeniería de Materiales, Facultad de Ingeniería, Universidad del Valle, Calle 13 No. 100-00, Santiago de Cali 760032, Colombia; sara.zamora@correounivalle.edu.co (S.I.Z.L.); Jefferson.murillo@correounivalle.edu.co (J.M.S.); valencia.mayra@correounivalle.edu.co (M.E.V.Z.); jose.mina@correounivalle.edu.co (J.H.M.H.); 2Programa de Química, Facultad de Ciencias, Universidad del Atlántico, Carrera 30 número 8-49, Puerto Colombia 081008, Colombia

**Keywords:** chitosan, spheres, morphology, physical, crosslinking, biomedical, application

## Abstract

Chitosan (CS) has special properties such as biocompatibility, biodegradability, antibacterial, and biological activity which make this material is currently studied in various applications, including tissue engineering. There are different methods to modify the morphology of CS. Most use chemical crosslinking agents, however, those methods have disadvantages such as low polymer degradability and unwanted side effects. The objective of this research was to obtain CS spheres through the physical crosslinking of commercial CS without using crosslinking agents through a simple coacervation method. A central composite experimental design was used to optimize the synthesis of the CS spheres and by the response surface methodology it was possible to obtain CS spheres with the smallest diameter and the most regular morphology. With the optimal formulation (CS solution 1.8% (*w*/*v*), acetic acid (AAC) solution 1% (*w*/*v*), sodium hydroxide (NaOH) solution 13% (*w*/*v*), relative humidity of (10%) and needle diameter of 0.6 mm), a final sphere diameter of 1 mm was obtained. Spheres were characterized by physical, chemical, thermal, and biological properties in simulated body fluid (SBF). The results obtained allowed us to understand the effect of the studied variables on the spheres’ diameter. An optimized condition facilitated the change in the morphology of the CS while maintaining its desirable properties for use in tissue engineering.

## 1. Introduction

Chitosan (CS) is a partially deacetylated derivative of chitin with biodegradation, biocompatibility, and bioactivity properties that make it attractive for biomedical applications (i.e., as a bioactive agent, scaffold manufacturing, hydrogels) and various dental applications [1].

CS can be prepared with different morphologies such as sponges, fibers, films, and other complex structures such as microspheres [2]. CS spheres are widely applied in many fields because of their special sizes and different forms and composition. CS spheres can be functionalized or crosslinked with chemical agents, either to modify their chemical properties such as the addition of active components like antibiotics, curcumin or zinc oxide, or simply to maintain their initial morphology such as the addition of crosslinking agents like glutaraldehyde or sodium tripolyphosphate [3,4,5,6,7].

Crosslinking mainly improves mechanical properties and increases polymer stability by interconnecting molecules. However, it simultaneously reduces the number of available functional groups, decreases polymer degradability, changes rheology properties [8]. Some crosslinking agents, like glutaraldehyde, are considered highly toxic. Due to its low molecular weight, glutaraldehyde can enter a living organism’s cells through several pathways [9,10]. Therefore, the safety of the use of chemical crosslinking is a fact that limits its application in the biomedical field [11].

CS spheres are generally used as a drug delivery system, quick hemostat system, tissue engineering as a microstructural unit in scaffolds, or active bone cement [7,12,13,14].

Studies report significant advantages when using CS with sphere morphology in different tissue engineering applications. For example, Abdel-Fattah et al. reported that it was possible to develop matrices with various pore size, controllable pore volume and excellent mechanical properties using CS spheres [15]. Zamora et al. concluded that the use of CS spheres as fillers in acrylic bone types of cement represents a significant improvement in mechanical properties compared to the use of CS flakes [16]. In their study, Meng et al. reported that when adding CS spheres in a bone cement formulation, they obtained a material with good bioactivity and biodegradation properties and a mechanical strength comparable to the reported strength of cancellous bone [14]. The objectives of CS sphere uses in tissue engineering, is mainly related to the biodegradation and biocompatibility properties of the material. Besides, CS spheres use is connected to controlling the surface area in contact with the fluid and increase or preserve the matrix’s mechanical properties.

Therefore, once the CS is in contact with the biological fluid, it is desirable to provide all the physical activity described before, maintaining its biocompatibility properties. The use of chemical crosslinking agents could be a barrier for CS to play its role. It can decrease CS degradability, and the crosslinking agent can cause unwanted side effects such as negative inside the body. Some authors have used physical crosslinking to avoid undesirable changes and possible side effects of chemical crosslinking [17]. Physical crosslinking generally occurs in response to specific conditions, like temperature or pH, and can be affected by polymer concentration and other components. CS physical crosslinking is problematic due to the nature of the molecule. Many interactions are involved at the same time, affecting the sol-gel transition [8].

Although there are investigations related to CS spheres prepared by different methods, in this study, we focus on the physical crosslinking of CS spheres by the simple coacervation method, looking for a statistical model that allows obtaining CS spheres with a small diameter and regular morphology.

A central composite experimental design was developed and allowed modeling and studying the effect of the process variables on the selected response variable, CS sphere diameter. The relation between the CS spheres formation solutions’ concentration, the drip needle’s size, and relative humidity for the CS spheres drying was statistically studied. In this way, it was possible to minimize the CS spheres’ size through the RSM. The optimization will allow for modifying the values of the factors conveniently according to the need and applying the proposed CS spheres. Finally, the CS spheres with the most regular morphology and smallest size possible demonstrated by scanning electron microscopy (SEM) were characterized and it was observed that by the proposed method of synthesis of CS spheres, CS presented a chemical modification, specifically in its degree of deacetylation, studied by elemental analysis (EA). The thermal analysis made it possible to verify that the CS spheres are less thermally stable. Finally, by the hydrolytic degradation test by immersion in simulated body fluid (SBF), it was studied the variations in pH of the medium, and it was proven that despite the physical and chemical modification of CS when synthesizing the spheres, they still present interesting degradation properties, characteristic of CS. Therefore, the CS spheres obtained show a regular morphology and suitable size and properties for tissue engineering applications.

## 2. Materials and Methods

The CS flakes employed in this study were from shrimp shells (*M*_w_ = 190,000–310,000 Da, viscosity of 328 CPS, and a deacetylation degree of 83%) according to the technical data from Sigma-Aldrich (St. Louis, MO, USA). For CS spheres synthesis, the last CS flakes were used, acetic acid (AAC) (*M*_w_ = 60.05 Da), and sodium hydroxide flakes (NaOH) (*M*_w_ = 40.00 Da), all also from Sigma-Aldrich.

### 2.1. Response Surface Design

This study’s experimental formulations were designed based on a central composite experimental design (Appendix A). In this case, the components’ effect, needle diameter, and the drying relative humidity (R.H.) in CS spheres’ final diameter were studied. Three independent central composite experimental designs were included for each R.H. studied (10%, 52%, room R.H.). The following components were used for experimental design: AAC, CS, NaOH. Intervals for the experimental region of each model are given in Table 1. The diameter is the response variable for each experimental design (Table 1).

The response surface methodology (RSM) was used to optimize the CS spheres’ diameter as the response variable using the prediction equation (Equation (1)):(1)$$Y=\;B_{o}+B_{1}X_{1}+B_{2}X_{2}+\cdots +B_{3}X_{1}^{2}+B_{4}X_{2}^{2}+\cdots +B_{5}X_{1}X_{2}+\cdots +e$$
where *Y* is the response variable (diameter), $B_{i}$ represent the model coefficients for predictive variables (i = 0, 1, 2, 3, 4, 5, … 10), $X_{\left(s\right)}$ are the independent variables (s = 1, 2, … 4), $X_{\left(s\right)}^{2}$ are the quadratic effects of each of the predictive variables, $X_{\left(s\right)}X_{\left(s\right)}$ are the interaction effect between variables, and e represents the experimental error.

### 2.2. CS Spheres Synthesis

Synthesis of CS spheres was performed using the coacervation methodology, which consisted in the addition of a low concentration (1% *w*/*v*) solution of CS flakes dissolved in diluted AAC (1.8% *w*/*v*) to a NaOH solution (13% *w*/*v*) dropwise, under gentle stirring.

#### 2.2.1. Dripping Stage

The assembly shown in Figure 1 was performed, which consisted of connecting a needle to a programmable syringe pump through which the CS solution passed and was deposited by dripping at a rate of 0.3 mL/min in the NaOH solution stirred at 360 rpm with the help of a magnetic stir plate.

Subsequently, the spheres were collected by filtration. Simultaneously, washes with 500 mL of distilled water at 360 rpm were carried out until a nearly neutral pH was obtained (6.8–7.5) using an Accumet AB150 pH meter (Fisher Scientific, Nepean, ON, Canada). Finally, the spheres were collected by filtration.

#### 2.2.2. Drying Stage

The CS spheres produced were dried for 24 h in desiccators (R.H. 10.0 ± 0.1%). Once the CS spheres were dried, they were collected to perform the granulometric separation with a 0.75 mm opening screen. Finally, they were collected in polyethylene bags and kept inside the desiccator at R.H. 10.0 ± 0.1%.

### 2.3. CS Spheres Characterization

#### 2.3.1. Particle Morphology and Surface Microstructure

Morphology of CS flakes and spheres were studied by scanning electron microscopy (SEM, JEOL JSM-6490LA, Musashino, Tokyo, Japan). Samples were gold-sputtered before the examination. The working conditions were 20 kV as acceleration voltage and mode of secondary backscattered electrons.

#### 2.3.2. Fourier-Transform Infrared Spectroscopy (FTIR)

The chemical identification of the CS flakes and spheres was studied using FTIR in the attenuated total reflectance (ATR) mode (Shimadzu, Kyoto, Japan).

#### 2.3.3. Elemental Analysis

The degree of deacetylation (DDA) of CS was also determined by elemental analysis. A Flash EA 1112 elemental analyzer (Thermo Scientific, Bremen, Germany) was used. The test was performed at 40 psi, with a constant flow of helium and oxygen. The DDA was calculated using Equation (2) [18]:(2)$$DDA\left(\%\right)=100-\left(\frac{\frac{C}{N}-5.2}{6.9-5.2}\right)\times 100\;\;\;\;\;$$

#### 2.3.4. Proton Nuclear Magnetic Resonance (^1^H-NMR)

For the ^1^H-NMR technique, an AVANCE II 400 spectrometer (Bruker, Billerica, MA, USA) was used. The degree of deacetylation of CS was determined. For this test, $D_{2}O$ was used with two drops of trifluoroacetic acid (reactive grade) as solvent and 3-(trimethylsilyl) propionic acid-d4, (reference salt), 32 runs were taken with an acquisition time of 5.24 s and a relaxation time of 1 s. Equation (3) was used to find the DDA [19]:(3)$$DDA\left(\%\right)=\left(1-\left(\frac{1}{3}HAc/\frac{1}{6}H26\right)\right)\times 100\;\;\;\;\;\;\;\;$$
where: *HAc* represents the three protons of the methyl group of N-acetyl glucosamine and *H*26 represents the protons from 2 to 6 ppm.

#### 2.3.5. Thermal Characterization

##### Differential Scanning Calorimetry (DSC)

The DSC technique was used to study the thermal behavior. The method was performed with a DSC25 instrument (TA Instruments, New Castle, DE, USA). Each sample was placed on an aluminum tray, and one was sealed with an empty tray of the same type (reference). A nitrogen atmosphere was used. The heating rate was 20 °C/min, and the temperature scans were taken from −100 °C to 550 °C. The second scanning was studied.

##### Thermogravimetric Analysis (TGA)

The thermal degradation of the CS spheres and its maximum decomposition temperature was studied using the TGA and the derivative of the thermogravimetric analysis (DTGA) technique. The technique was performed with a TA Instruments TGAQ50 system. A nitrogen atmosphere was used. The heating rate used was 20 °C/min, and the temperature ranges were brought from room temperature to 550 °C.

#### 2.3.6. Hydrolytic Degradation Assay

CS spheres immersion for 30 days in simulated body fluid (SBF) prepared according to Kokubo and Takadama method was used to evaluate the hydrolytic degradation [20]. SBF was stored in plastic bottles at a temperature between 5 and 10 °C and used within less than one month. During immersion, the pH of the sample solution was analyzed. Each sample (0.12 g) was immersed in SBF and kept in an incubator at 37 °C for 30 days in which pH measurements were made daily.

### 2.4. Statistical Analysis and Optimization

The design experiment optimization included analyzing each optimization graph, and contour plots provided by the MINITAB 18 program (Minitab, State College, PA, USA,) in the central composite experiment design for each R.H. For the optimization, the desirability function (D: global; d: individual) was used, which combines the geometric media and optimizing the general metric media [21].

## 3. Results

Through the development of the experimental design, it was observed that the CS/AAC solution should have a viscosity that would allow it to pass through the drip needle without effort to produce a spherical drop. If the dissolution had an excessive viscosity, it could not flow through the drip needle. In contrast, if the solution viscosity is too low, it would not produce spherical drops of CS, and instead, it will create a continuous stream of solution. It was observed that NaOH concentration must be enough to able CS/AAC solution to form regular spheres upon contact with NaOH solution. Furthermore, with a controlled drying R.H., spheres with a more regular morphology were obtained.

The central composite experimental design allowed modeling the response variable (CS sphere diameter) with precision, using a second-degree polynomial (Equation (1)), through the analysis of the interaction between the variables. Statistical models for each condition were obtained (Appendix A). Equation (4) corresponds to the statistical model for the conditions R.H. 10% and needle 1:(4)$$\begin{array}{ll} Diameter\;= & 15,82\;+\;16,4\;AAC\;-\;1239\;CS\;-\;67,8\;NaOH\;-\;769\;AAC\times AAC\\ & \;+\;33381\;CS\times CS+\;265,1\;NaOH\times NaOH\;+\;3024\;AAC\\ & \;\times CS \end{array}$$

The Pareto charts of the standardized effects (Figure 2, Appendix A) graphically represent the magnitude of the process variables’ effects on the response variable within the proposed design. The absolute value of the standardized effects can be observed, and any effect that extends beyond the reference line is statistically significant. According to the Pareto chart shown in Figure 2, the linear effect and the quadratic effects and interactions between the process variables are significant. These effects effectively predict the CS sphere diameter. The model allowed the verification of the relationship that the process variables have with the response variable studied and generated a more straightforward concept of the proposed method for the CS spheres synthesis. In general, for the three environments, the models’ adjustments were outstanding, and therefore, this model was used to optimize the diameter variable.

The response surface methodology (RSM) is a technique that optimizes one or more response variables within an experimentation area. RSM was used to optimize the response variable and obtain the minimum CS spheres diameter depending on the process variables. The optimization of the spheres’ diameter by RSM allowed obtaining a diameter of CS spheres within the expected range.

The contour graphs allowed us to observe the different CS spheres diameters according to the conditions of the independent variables in each R.H. (Appendix A). According to the contour graphs shown in Figure 3, the smallest diameters of the spheres (<1 mm) were obtained with the presence of the highest percentages of CS, low concentration of AAC, and low concentrations of NaOH. Further, the optimization charts (Appendix A) offer the possibility to modify the values of the factors conveniently. This design optimization allowed finding the most appropriate conditions to minimize the diameter and obtain a homogeneous morphology in the spheres; these conditions are shown in Table 2. 

### 3.1. Particle Morphology and Surface Microstructure

CS spheres obtained were analyzed using a stereoscope at magnification 80×, using a graph paper as a reference frame to study the variations in the morphology of CS spheres. Sphere morphology was obtained with a diameter of 1 mm or lower, as shown in Figure 4.

Furthermore, SEM was performed on the commercial CS flakes and CS spheres. As can be seen in Figure 5, commercial CS has an irregular laminar morphology and angular edges. On the other hand, the coacervation mechanism’s CS particles have an almost spherical morphology with a mainly smooth surface, but some defects are introduced after the manual separation process.

### 3.2. Fourier-Transform Infrared Spectroscopy (FTIR)

Chemical analysis was carried out using FTIR. Figure 6 shows the FTIR spectra of CS flakes and spheres. Both samples showed prominent characteristic bands. Overlapping bands at 3286 cm^−1^, correspond to the stretching vibration of O-H and NH_2_ groups. The bands present at 2915 cm^−1^ and 2875 cm^−1^ are associated with NH_2_ and CH_2_ groups. The band presents at 1643 cm^−1^ corresponds to the NH_2_ (Amide I) group’s bending, while the band at 1566 cm^−1^ corresponds to NH_2_ (Amide II). The bands at 1367 cm^−1^ and 1306 cm^−1^ correspond to CH bending and C-CH_3_ deformation, respectively. Finally, at 1020 cm^−1^, a band is associated with the glucopyranose ring [22,23,24]. The physical crosslinking performed to synthesize the chitosan spheres did not significantly affect the chemical structure of the polymer. Both spectra show the characteristic bands of CS with a slight variation in intensity. Studies report that variations in the intensity of the peaks are associated with the higher presence of specific functional groups, related to the variation of DDA [25,26].

### 3.3. Elemental Analysis

According to the elemental analysis (Table 3), the percentage of the elements are similar in flakes and CS spheres, as expected. However, nitrogen content presents some differences, probably due to the amine group interaction with sodium ions from sodium hydroxide interaction. A similar process occurs in alkaline deacetylation with the acetamide groups of chitin [27]. Therefore, the DDA values (Equation (2)) by EA were 88.43% and 58.94% for CS flakes and spheres, respectively, and by ^1^H-NMR (Equation (3)), the DDA values obtained were of 75.01% and 60.78% (Appendix A).

According to different studies, a higher DDA (%) increases the number of reactive positions for degradation by hydrolysis and provides a lower crystallinity of the polymer, which makes the polymer’s degradation rate higher [28,29,30]. Furthermore, CS’s degradation capacity influences the solubility and mechanical properties due to the structural changes that the material undergoes [31]. Consequently, this property will control facile CS degradation as required in biomedical applications.

### 3.4. Thermal Characterization

#### 3.4.1. Differential Scanning Calorimetry (DSC)

Figure 7 shows the DSC curves of CS flakes and CS spheres. The first endothermic event between room temperature and 150 °C corresponds to the loss of absorbed and constituent water molecules, volatile components, and hydrophilic groups present in CS. 

Around 156 °C, an additional endothermic peak is observed, associated with the polymer’s glass transition temperature. They coincided with Dong et al., who reported a glass transition temperature for CS between 140 °C and 150 °C. Dong et al. also reported a thermal transition at 200 °C associated with the liquid-liquid change, which is also present in both analyzes for flakes and spheres at approximately 220 °C [32]. The exothermic events at 320 °C and 290 °C for CS flakes and spheres are associated with the decomposition of amine units, cracking, and CS [24,33].

#### 3.4.2. Thermogravimetric Analysis (TGA)

Figure 8a shows the TGA curves of CS flakes and spheres. The first stage of degradation occurs between 35 °C to 200 °C, with a weight loss of approximately 12%, related to the evaporation of water molecules or small molecules [34]. According to the curve, these elements’ weight loss occurs more pronounced in the CS flakes than the CS spheres.

This can be explained by the physical crosslinking when preparing the CS spheres, which reduces the number of available functional groups, limiting the interaction with water molecules present in the environment [35].

The second stage of degradation occurs between 200 °C to 550 °C, with a weight loss of 68% and 74%, corresponding to CS’s flakes and spheres, respectively. This stage is related to a complex process that involves the dehydration of the saccharide rings and the decomposition of the acetylated and deacetylated units of CS [34]. Furthermore, in Figure 8b, in the DTGA curves, it can be seen that CS spheres present a first peak representing the maximum rate of degradation, which occurs at 290 °C (decomposition temperature). Then another small peak is observed at 307 °C, while CS flakes’ decomposition temperature is 309 °C. Gámiz-González et al. observed similar behavior when analyzing the TGA and DTGA of chitosan and its acetylated derivatives with different DDA. In the DTGA, they reported a second peak’s appearance as a shoulder on the high-temperature side of the principal peak, concluding that the intensity of this second peak increased as the DDA decreased [36]. According to Young Sik et al., the first decomposition stage corresponds to d-glucosamine chitosan units’ depolymerization. In contrast, the second decomposition stage can be attributed to N-acetyl-d-glucosamine [37].

In this study, the peak corresponding to the degradation of N-acetyl-d-glucosamine is present with greater intensity. This is due to the high proportion of these units within the CS spheres’ polymer chain compared with CS flakes.

Therefore, from the thermal analyses it can be concluded that the CS spheres are less stable thermally than the CS flakes. Some studies indicate that thermal stability of CS is linked to the degree of deacetylation, as a consequence of the decrease in the content of acetyl groups, which provides less crystallinity to CS, and reduces thermal stability because less energy is required for its decomposition [24,34,38]. Takara et al. reported that treating CS with aqueous NaOH solutions causes deprotonation, new hydrogen bonds, and greater deacetylation of CS [39].

### 3.5. Hydrolytic Degradation Assay

Figure 9 shows the medium’s pH variation during the immersion of the CS flakes and CS spheres inside the SBF. The medium’s pH tends to decrease gradually with days, which might be due to the CS degradation products mainly composed of bound 1,4-d-glucosamine and partially -1,4 N-acetyl-d-glucosamine. Studies report that these products have significant potential in medical fields due to their wide bioactivity, antibacterial, antifungal, antitumor activity, radical scavenging, antimicrobial activity, immunity, modulating effect, and wound healing [40].

The pH values of SBF with CS flakes were higher than those with spheres until day 26th. On day 26th, the value of 6.8 for flakes and 5.9 with spheres was observed. The test runs until day 30th. There was no variation in pH, with a final value of 5.6 and 5.7 for flakes and spheres, respectively. The values were too acid for biomedical applications according to the traditional standards [41].

For biomedical applications, the CS compounds should maintain the biodegradation and biocompatibility properties. Therefore, when observing that the physical crosslinking performed for the synthesis of the CS spheres did not have a significant effect on the performance of the spheres within the hydrolytic degradation assay is positive for the research since this means that it was possible to transform the morphology of CS flakes into a regular, spherical morphology without considerably modifying the chemical and biological properties of CS in flakes. Then, the use of CS spheres will be feasible in biomedical applications, where it is required to use CS due to its excellent biological and chemical properties, and also, the morphology of the spheres represents advantages within the application.

For example, in a previous study, the improvement in CS spheres’ mechanical performance was confirmed compared with CS flakes within an acrylic bone cement matrix [16]. This is a desirable performance in other applications such as scaffolds or films or matrices for tissue engineering. Kucharska et al., in their study, manufactured scaffolds of agglomerated CS spheres for application in tissue engineering. The spheres were synthesized with different crosslinking agents and had a diameter of approximately 1 mm [7]. Shu et al. reported using ionically crosslinked CS spheres of 1 mm of diameter, with application in the administration and release of specific drugs within the stomach [6]. This is a potential application for the produced CS spheres by adding dispersed components as model drugs to the CS-AAC solution and performing the proposed synthesis method. Other authors report smaller diameter CS spheres using different crosslinking agents for different medical applications [13,42,43,44].

In Figure 10, CS spheres’ surface morphology presented modifications with the immersion time in the SBF. Initially, spheres offer a regular edge and a smooth surface, but from day 5th^,^ the surface increases the roughness and porosity, which is very convenient for tissue engineering applications. Porous structures’ formation is acceptable because porosity structures mimic the natural extracellular matrix better than a two-dimensional system like CS flakes, supporting adhesion and cell growth [45].

## 4. Conclusions

The coacervation methodology successfully produced CS spheres of a diameter of 1.0 ± 0.1 mm with a consistent and homogeneous appearance by varying the solutions’ concentrations, the drip needle’s diameter, and the relative humidity during drying, demonstrating the reproducibility of the method.

CS sphere deacetylation degree characterization confirmed the physical crosslinking by decreasing the value compared to the CS flakes. The lower hydrogen bonding capacity of the CS spheres also produced a decrease in the maximum rate of degradation, as evidenced by TGA and DSC analysis (309° and 290° for CS flakes and spheres, respectively), demonstrating that physical crosslinking decreases CS crystallinity by reducing the number of available hydrogen bonds. When CS spheres were evaluated within the SBF, they presented a similar behavior regarding the final pH reported during the test, 6.8 for CS flakes and 5.9 for CS spheres, reporting values allowed by the standard. The synthesis method generated a physical crosslinking that did not modify CS’s biological properties; therefore, CS spheres will have remarkably similar properties to CS flakes in terms of degradation and biocompatibility. Furthermore, after the hydrolytic assay, SEM analysis demonstrated that immersion produced a porous structure, which means that the CS spheres will have more significant degradation properties because this structure will help the tissue promoter cells enter the system. Moreover, it will also facilitate the transfer of oxygen, nutrients, and metabolic waste due it mimics the extracellular matrix.

## Figures and Tables

**Figure 1 biomimetics-05-00063-f001:**
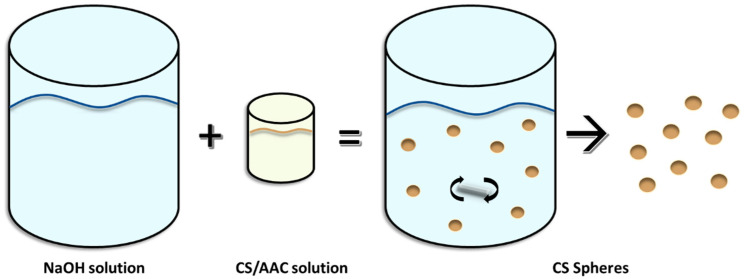
Simple coacervation method for CS spheres synthesis.

**Figure 2 biomimetics-05-00063-f002:**
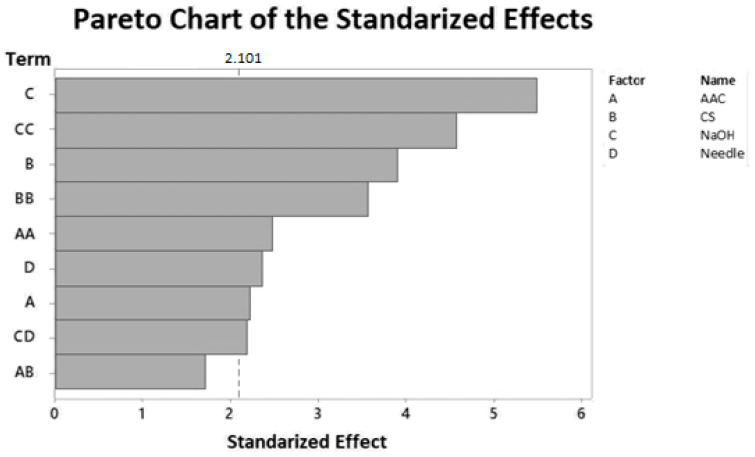
Pareto chart of the standardized effects. Response: Diameter. R.H. 10% ± 0.1%.

**Figure 3 biomimetics-05-00063-f003:**
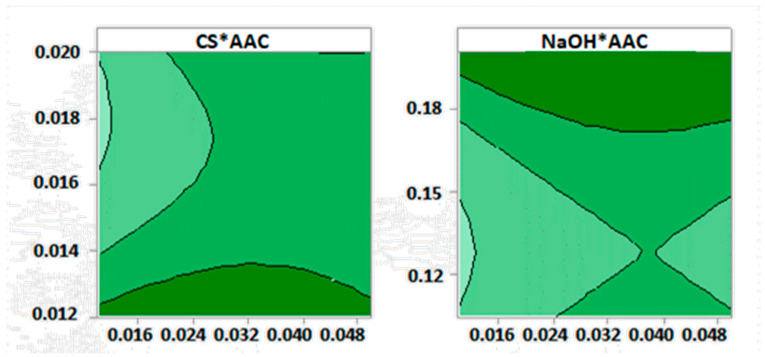
Diameter contour plot for CS spheres. Conditions: Needle 1, R.H. 10% ± 0.1%.

**Figure 4 biomimetics-05-00063-f004:**
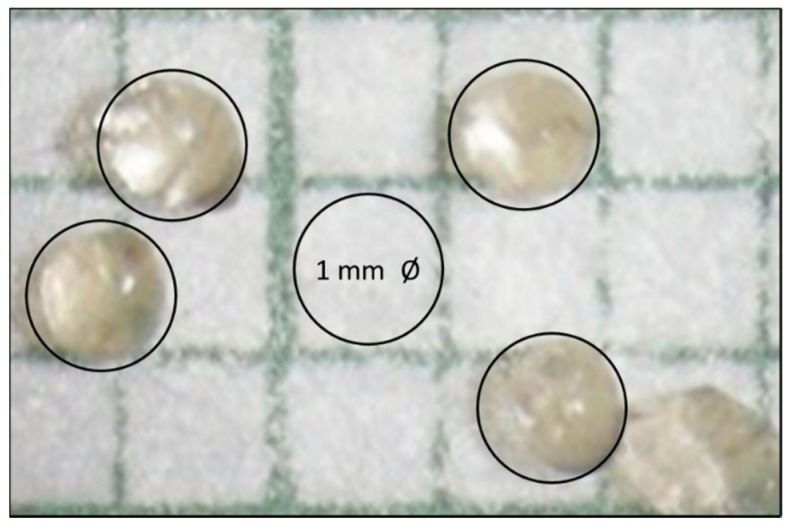
The photograph at 80× of CS spheres obtained from the optimal formulation (R.H. 10%).

**Figure 5 biomimetics-05-00063-f005:**
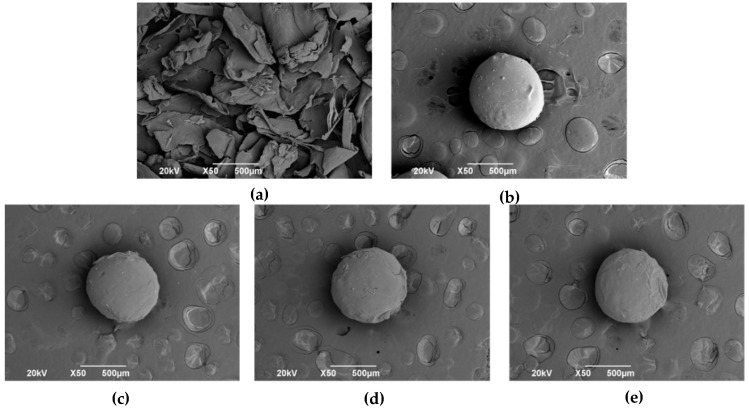
SEM micrographs at 50× of: (**a**) CS flakes (**b**–**e**) CS spheres [16].

**Figure 6 biomimetics-05-00063-f006:**
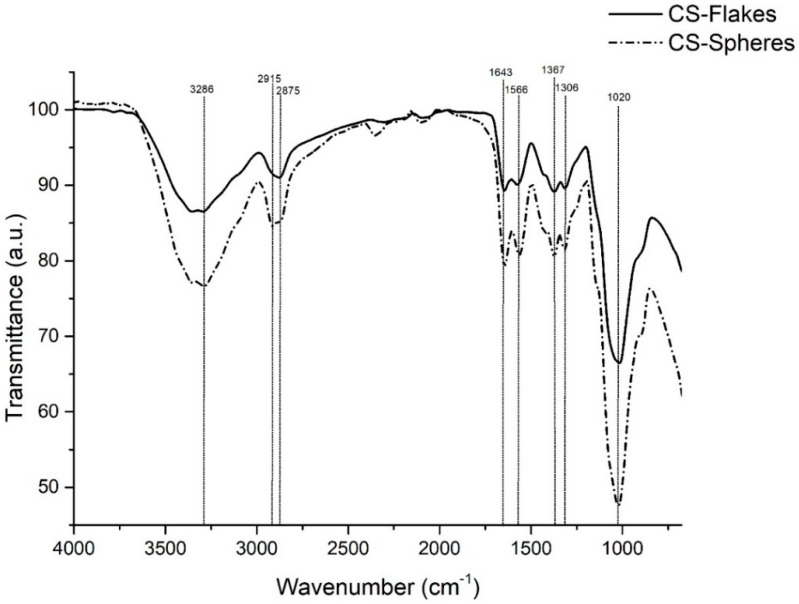
FTIR spectra of CS flakes and CS spheres.

**Figure 7 biomimetics-05-00063-f007:**
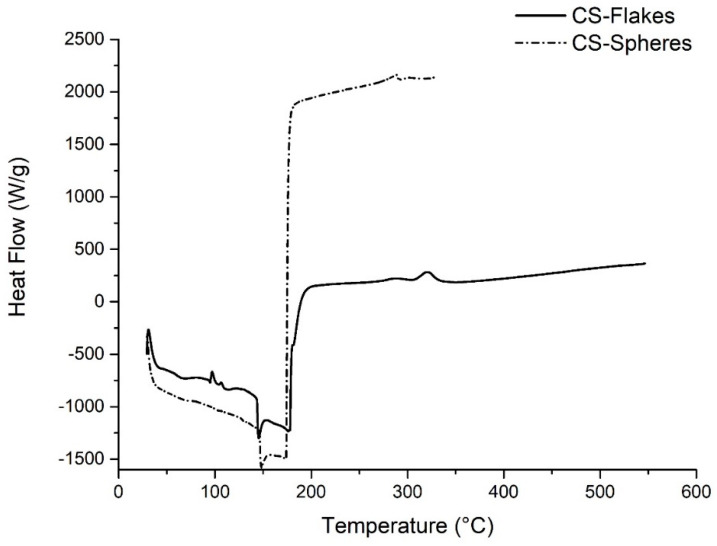
DSC analysis for CS flakes and spheres.

**Figure 8 biomimetics-05-00063-f008:**
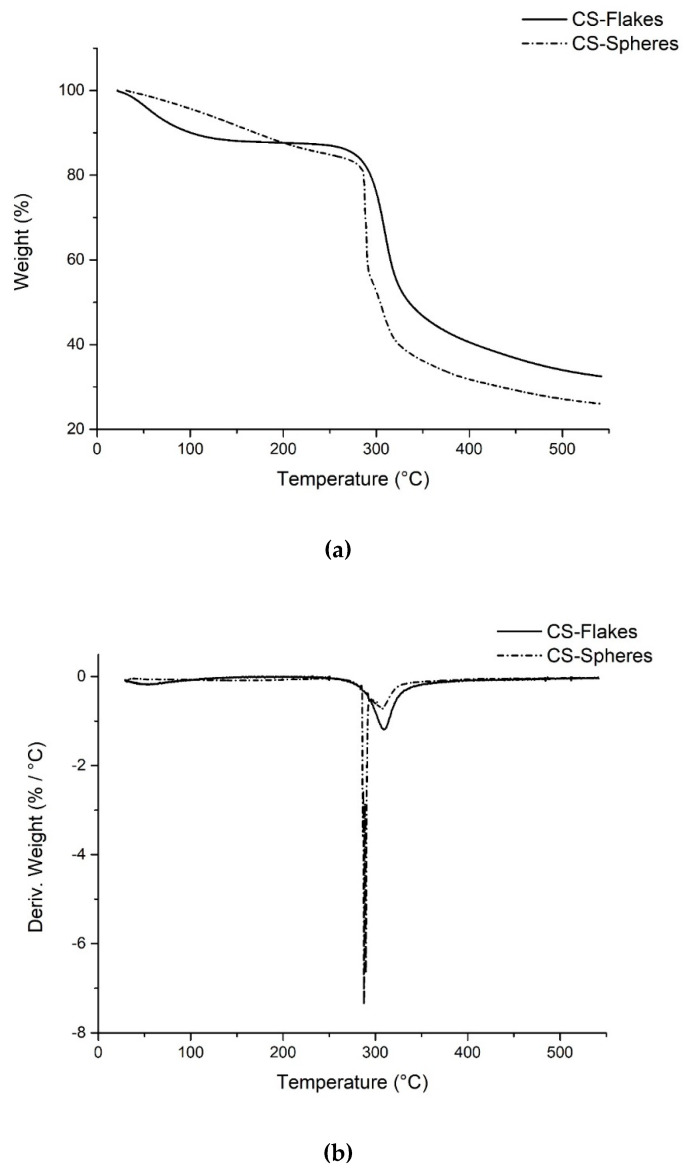
Thermal analysis of CS flakes and spheres: (**a**) TGA curves (**b**) DTGA curves.

**Figure 9 biomimetics-05-00063-f009:**
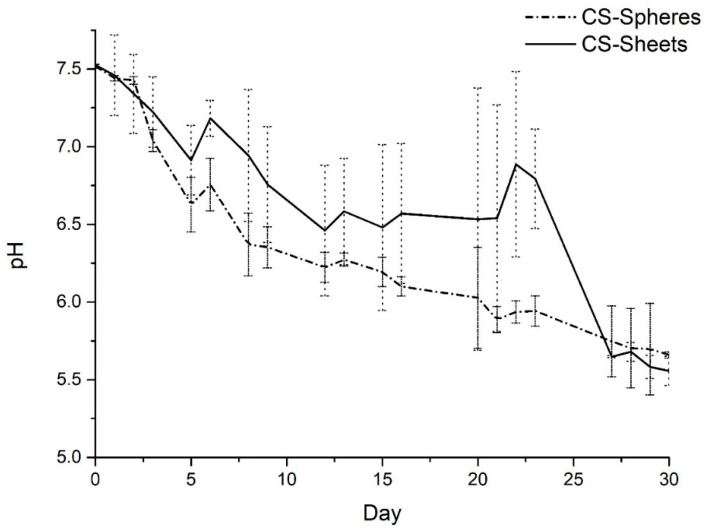
pH changes in CS flakes and CS spheres after 30 days of immersion in SBF.

**Figure 10 biomimetics-05-00063-f010:**
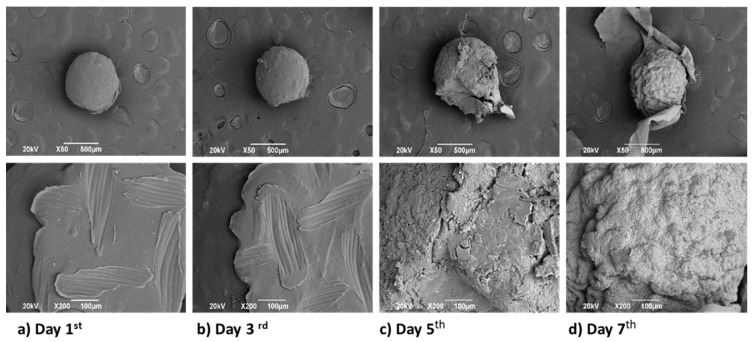
SEM micrographs at 50× and 200× of CS spheres from day 1st until day 28th of SBF immersion. (**a**) Day 1^st^, (**b**) Day 3^rd^, (**c**) Day 5^th^, (**d**) Day 7^th^, (**e**) Day 14^th^, (**f**) Day 21^st^, (**g**) Day 28^th^.

**Table 1 biomimetics-05-00063-t001:** Independent variables of the experimental design for the CS spheres synthesis.

Independent Variables	Intervals		
	Minimum (% g/mL)	Centered (% g/mL)	Maximum (% g/mL)
$x_{1}$ Acid acetic concentration (AAC)	1	3	5
$x_{2}$ CS concentration (CS)	1.2	1.7	2
$x_{3}$ Sodium hydroxide concentration (NaOH)	10.4	1.4	20
$x_{4}$ Needle diameter	0.6 (mm)		0.8 (mm)

**Table 2 biomimetics-05-00063-t002:** The optimal formulation for CS spheres synthesis.

Component	Concentration (% *w*/*v*)
AAC	1
CS	1.8
NaOH	13

**Table 3 biomimetics-05-00063-t003:** Elemental analysis for CS flakes and CS spheres.

Element	CS Flakes	CS Spheres
	(%)	(%)
Nitrogen	7.28	6.39
Carbon	38.93	37.38
Hydrogen	6.98	7.04
Sulfur	0	0
Total	53.20	50.82
DDA	88.43	58.94

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
