# Peer review of "Optimization by Central Composite Experimental Design of the Synthesis of Physically Crosslinked Chitosan Spheres"

_biomimetics, 2020, doi:10.3390/biomimetics5040063_

Round 1

Reviewer 1 Report

  1. There are already numerous publications to produce Crosslinked chitosan microspheres.  I do not see any advantage of your method over those many other methods.  I wish you had reviewed all the previous ones appropriately.
  2. I do not see the need to use colors in plotting the few figures.  They can be in black and white.
  3.  Such a simple experiment and so many co-authors for that!!  I really do not see the roles of so many co-authors for this simple research

Author Response

Reviewer 1

Comments and Suggestions for Authors

  1. There are already numerous publications to produce Crosslinked chitosan microspheres.  I do not see any advantage of your method over those many other methods.  I wish you had reviewed all the previous ones appropriately.

R// We appreciate the reviewer's comment. About the advantage of the paper, we agree that several papers have been reported about chitosan microspheres. However, most based on chemical crosslinking of the spheres[1–4], generating undesirable effects on the spheres: toxicity [5,6], reduction in the number of available functional groups, decreased polymer degradability, and changes rheology properties [7].

Few studies report only physical crosslinking is used [8,9], while the combined effects of factors that could affect particle size are not reported.

CS spheres obtained with physical crosslinking have properties similar to CS sheets, particularly their chemical composition and degradability. Therefore, the advantage of this synthesis method is that the CS spheres produced can be used in the same applications where CS sheets are usually used, with the advantage that the CS spheres have better mechanical properties and stability, also to have the ability to form porous structures and to control particle size according to the specific need of the application.

Given the significant interest that microspheres have in biomedical applications, the number of papers published on the subject increases every day [10–13]. Our research's contribution consisted of optimizing the process variables to different relative humidity through an experimental design that allowed quantifying their contribution to the spheres' final diameter. The knowledge of the effect of the process variables serves as a base in future investigations to adjust conditions and predict and obtain the final diameters required according to the need and specific application [14–19].

  1. I do not see the need to use colors in plotting the few figures.  They can be in black and white

R// We appreciate the reviewer's comment. Figures 2, 4, 6, 7, 8, and 9 are now without color (black and white).

  1. Such a simple experiment and so many co-authors for that!!  I really do not see the roles of so many co-authors for this simple research

R// We appreciate the reviewer's comment. We can clarify that this paper is part of a broader research project in which the spheres generated are explored in different biomedical applications [19]. Students and senior researchers contribute to the interdisciplinary research group, and their contribution is found at the end of the text in the authors' contribution section.

References

  1. Wang, Y.M.; Sato, H.; Adachi, I.; Horikoshi, I. Optimization of the Formulation Design of Chitosan Microspheres Containing Cisplatin. J. Pharm. Sci. 1996, 85, 1204–1210, doi:https://doi.org/10.1021/js960092j.
  2. Patel, M.M. Formulation and development of di-dependent microparticulate system for colon-specific drug delivery. Drug Deliv. Transl. Res. 2017, 7, 312–324, doi:10.1007/s13346-017-0358-7.
  3. Biró, E.; Németh, A.S.; Feczkó, T.; Tóth, J.; Sisak, C.; Gyenis, J. Three-step experimental design to determine the effect of process parameters on the size of chitosan microspheres. Chem. Eng. Process. Process Intensif. 2009, 48, 771–779, doi:https://doi.org/10.1016/j.cep.2008.10.001.
  4. Zhou, X.; Cheng, X.J.; Liu, W.F.; Li, J.; Ren, L.H.; Dang, Q.F.; Feng, C.; Chen, X.G. Optimization and characteristics of preparing chitosan microspheres using response surface methodology. J. Appl. Polym. Sci. 2013, 127, 4433–4439, doi:10.1002/app.38003.
  5. Gupta, K.C.; Jabrail, F.H. Effects of degree of deacetylation and crosslinking on physical characteristics, swelling and release behavior of chitosan microspheres. Carbohydr. Polym. 2006, 66, 43–54, doi:10.1016/j.carbpol.2006.02.019.
  6. Luo, K.; Yang, Y.; Shao, Z. Physically Crosslinked Biocompatible Silk-Fibroin-Based Hydrogels with High Mechanical Performance. Adv. Funct. Mater. 2016, 26, 872–880, doi:10.1002/adfm.201503450.
  7. Reddy, N.; Reddy, R.; Jiang, Q. Crosslinking biopolymers for biomedical applications. Trends Biotechnol. 2015, 33, 362–369.
  8. Dias, F.S.; Queiroz, D.C.; Nascimento, R.F.; Lima, M.B. Um sistema simples para preparação de microesferas de quitosana . Química Nov. 2008, 31, 160–163.
  9. Torres, M.A.; Vieira, R.S.; Beppu, M.M.; Santana, C.C. Production of chitosan microspheres by spraying and coagulation process. In Proceedings of the Transactions - 7th World Biomaterials Congress; School of Chemical Engineering, State University of Campinas, São Paulo, Brazil, 2004; p. 1516.
  10. Kim, J.U.; Shahbaz, H.M.; Lee, H.; Kim, T.; Yang, K.; Roh, Y.H.; Park, J. Optimization of phytic acid-crosslinked chitosan microspheres for oral insulin delivery using response surface methodology. Int. J. Pharm. 2020, 588, 119736, doi:https://doi.org/10.1016/j.ijpharm.2020.119736.
  11. Wang, S.; Sun, Y.; Zhang, J.; Cui, X.; Xu, Z.; Ding, D.; Zhao, L.; Li, W.; Zhang, W. Astragalus Polysaccharides/Chitosan Microspheres for Nasal Delivery: Preparation, Optimization, Characterization, and Pharmacodynamics . Front. Pharmacol.   2020, 11, 230.
  12. ERTEN TAYSI, A.; CEVHER, E.; SESSEVMEZ, M.; OLGAC, V.; MERT TAYSI, N.; ATALAY, B. The efficacy of sustained-release chitosan microspheres containing recombinant human parathyroid hormone on MRONJ . Brazilian Oral Res. 2019, 33.
  13. Jena, G.K.; Sahoo, S.K.; Patra, C.N.; Panigrahi, K.C.; Sahu, S.; Dixit, P.K. Design, optimization, and evaluation of capecitabine-loaded chitosan microspheres for colon targeting. Asian J. Pharm. 2017, 11, S592–S602.
  14. Shu, X..; Zhu, K.. Controlled drug release properties of ionically crosslinked chitosan beads: the influence of anion structure. Eur. J. Pharm. Biopharm. 2002, 54, 235–243, doi:10.1016/S0939-6411(02)00052-8.
  15. Abdel-fattah, W.I.; Jiang, T.; El-Bassyouni, G.E.-T.; Laurencin, C.T. Synthesis , characterization of chitosans and fabrication of sintered chitosan microsphere matrices for bone tissue engineering. Acta Biomater. 2007, 3, 503–514, doi:10.1016/j.actbio.2006.12.004.
  16. Li, J.; Wu, X.; Wu, Y.; Tang, Z.; Sun, X.; Pan, M.; Chen, Y.; Li, J.; Xiao, R.; Wang, Z.; et al. Porous chitosan microspheres for application as quick in vitro and in vivo hemostat. Mater. Sci. Eng. C 2017, 77, 411–419, doi:10.1016/j.msec.2017.03.276.
  17. Domalik-Pyzik, P.; ChĹ‚opek, J.; Pielichowska, K. Chitosan-Based Hydrogels: Preparation, Properties, and Applications. In; Springer, Cham, 2018; pp. 1–29.
  18. Meng, D.; Dong, L.; Wen, Y.; Xie, Q. Effects of adding resorbable chitosan microspheres to calcium phosphate cements for bone regeneration. Mater. Sci. Eng. C 2015, 47, 266–272, doi:10.1016/j.msec.2014.11.049.
  19. Zamora Lagos, S.I.; Murillo Salas, J.; Valencia Zapata, M.E.; Mina Hernandez, J.H.; Valencia, C.H.; Rojo, L.; Grande Tovar, C.D. Influence of the chitosan morphology on the properties of acrylic cements and their biocompatibility. RSC Adv. 2020, 10, 31156–31164, doi:10.1039/D0RA06508K.

Reviewer 2 Report

- Abstract is too long. Please rewrite this part of the manuscript. The abstract should state briefly the purpose of the research, the principal results and major conclusions.

-Lines 83-85. “Although there are different investigations related to CS spheres preparation using chemical crosslinking, in this study, CS spheres we focus on the physical crosslinking by the simple coacervation method but optimizing the formulation for CS spheres preparation.” Please rewrite this sentence and remove “chemical crosslinking”.

-Please explain why have you decided to use a central composite design? And why the optimization of the CS spheres preparation is so important?

-In the introduction the new aspect of the work should be highlighted, and the hypothesis should be given.

-The quality of the Fig. 1 should be improved.

- The obtained model should be presented .

Author Response

Reviewer 2

Comments and Suggestions for Authors

  1. Abstract is too long. Please rewrite this part of the manuscript. The abstract should state briefly the purpose of the research, the principal results and major conclusions.

R// We appreciate the reviewer's comment. The abstract was modified and shortened, including the reviewer's suggestion.

  1. Lines 83-85. "Although there are different investigations related to CS spheres preparation using chemical crosslinking, in this study, CS spheres we focus on the physical crosslinking by the simple coacervation method but optimizing the formulation for CS spheres preparation." Please rewrite this sentence and remove "chemical crosslinking."

R// We appreciate the reviewer's comment. We rewrote the sentence correctly.

  1. Please explain why have you decided to use a central composite design? And why the optimization of the CS spheres preparation is so important?

R// We appreciate the reviewer's comment. We used central composite design because it allowed us to estimate the diameter behavior within the range of the studied variables, optimizing the process conditions according to our specification of minimizing the spheres' diameter variable.

Given the significant interest that microspheres have in biomedical applications, the number of papers published on the subject increases every day [1 - 4]. Our research's contribution consisted of optimizing the process variables by using an experimental design that allowed quantifying their contribution to the spheres' final diameter. The mathematical model obtained to estimate the spheres' diameter serves as a basis in future investigations to adjust conditions and predict the final diameters required according to the need and specific application.

Lines 194-206 and 214-216 were added to the text to clarify the reviewer's suggestion.

  1. The new aspect of the work should be highlighted in the introduction, and the hypothesis should be given.

R// We appreciate the reviewer's comment. In the introduction, lines 75 – 83 were added.

  1. The quality of the Fig. 1 should be improved.

R// We appreciate the reviewer's comment. Figure 1 was modified to improve quality.

  1. The obtained model should be presented.

R// We appreciate the reviewer's comment. The mathematical model was presented in the text in lines 194-209. Additionally, an analysis of the Pareto chart was added.

References

  1. Kim, J.U.; Shahbaz, H.M.; Lee, H.; Kim, T.; Yang, K.; Roh, Y.H.; Park, J. Optimization of phytic acid-crosslinked chitosan microspheres for oral insulin delivery using response surface methodology. Int. J. Pharm. 2020, 588, 119736, doi:https://doi.org/10.1016/j.ijpharm.2020.119736.
  2. Wang, S.; Sun, Y.; Zhang, J.; Cui, X.; Xu, Z.; Ding, D.; Zhao, L.; Li, W.; Zhang, W. Astragalus Polysaccharides/Chitosan Microspheres for Nasal Delivery: Preparation, Optimization, Characterization, and Pharmacodynamics . Front. Pharmacol.   2020, 11, 230.
  3. Erten Taysi , A.; Cevher, E.; Sessevmez, M.; Olgac, V.; Mert Taysi, N.; Atalay, B. The efficacy of sustained-release chitosan microspheres containing recombinant human parathyroid hormone on MRONJ . Brazilian Oral Res. 2019, 33.
  4. Jena, G.K.; Sahoo, S.K.; Patra, C.N.; Panigrahi, K.C.; Sahu, S.; Dixit, P.K. Design, optimization, and evaluation of capecitabine-loaded chitosan microspheres for colon targeting. Asian J. Pharm. 2017, 11, S592–S602.

Reviewer 3 Report

In this article the authors performed the physical modification of comercial chitosan sheets to obtain chitosan spheres through physical crosslinking using the simple coacervation method. The authors found and optimized a simple way to obtain CS spheres without changing the biological properties of chitosan. Chitosan has very good properties that all we know, because it is very studied. The strategy used by the authors is interesting and importante, because they do not use crosslinking agentes, which in most cases they are toxic. It looks like everything is fine, but there are questions for the authors answer

  • What is the feasibility of these CS spheres be used in medical applications
  • With a diameter of 1mm in CS spheres, what could be the medical applications?
  • CS spheres are generally used as a drug delivery system, taking this into account, how do the functionalization of the CS spheres with active componentes?
  • The authors must presente images where more CS spheres can be seen and not just one CS spheres. An image with only one CS sphere is not representative

Author Response

Reviewer 3

Comments and Suggestions for Authors

  1. What is the feasibility of these CS spheres be used in medical application?

R// We appreciate the reviewer's question. These CS spheres have great potential for medical applications [1] due to their similar properties to CS sheets, particularly their chemical composition and degradability. Therefore, the advantage of this synthesis method is that the CS spheres produced can be used in the same applications where CS sheets are usually used, with the advantage that CS spheres have better mechanical properties and stability, besides, to have the ability to form porous structures and to control particle size according to the specific need of the application.

  1. With a diameter of 1mm in CS spheres, what could be the medical applications?

R// We appreciate the reviewer's question. Lines 338 - 348 in the text mention the following: For example, in a previous study, the improvement in CS spheres' mechanical performance was confirmed compared with CS flakes within an acrylic bone cement matrix [16]. This is a desirable performance in other applications such as scaffolds or films or matrices for tissue engineering. Kucharska et al., in their study, manufactured scaffolds of agglomerated CS spheres for application in tissue engineering. The spheres were synthesized with different crosslinking agents and had a diameter of approximately 1 mm [7]. Shu et al. reported using ionically crosslinked CS spheres of 1 mm of diameter, with application in the administration and release of specific drugs within the stomach [6].

  1. CS spheres are generally used as a drug delivery system, taking this into account, how do the functionalization of the CS spheres with active components?

R// We appreciate the reviewer's question. Some additional information was added in Line 348-351: This is a potential application for the produced CS spheres by adding dispersed components as model drugs to the CS-AAC solution and performing the proposed synthesis method. Other authors report smaller diameter CS spheres using different crosslinking agents for different medical applications [13,42–44].

  1. The authors must present images where more CS spheres can be seen and not just one CS spheres.

R// We appreciate the reviewer's comment. More SEM images of different CS spheres were added to Figure 4.

References

  1. Zamora Lagos, S.I.; Murillo Salas, J.; Valencia Zapata, M.E.; Mina Hernandez, J.H.; Valencia, C.H.; Rojo, L.; Grande Tovar, C.D. Influence of the chitosan morphology on the properties of acrylic cements and their biocompatibility. RSC Adv. 2020, 10, 31156–31164, doi:10.1039/D0RA06508K.

Round 2

Reviewer 2 Report

Many thanks to the authors for their comprehensive responses to my comments. The paper is ready to be published in the journal.

Reviewer 3 Report

The revised manuscript is ok for me